# Changes in Alcohol Consumption and Risk of Heart Failure: A Nationwide Population-Based Study in Korea

**DOI:** 10.3390/ijerph192316265

**Published:** 2022-12-05

**Authors:** Yohwan Yeo, Su-Min Jeong, Dong Wook Shin, Kyungdo Han, Juhwan Yoo, Jung Eun Yoo, Seung-Pyo Lee

**Affiliations:** 1Department of Family Medicine, College of Medicine, Hallym University Dongtan Sacred Heart Hospital, Hwaseong 18450, Republic of Korea; 2Department of Preventive Medicine, Seoul National University College of Medicine, Seoul 03080, Republic of Korea; 3Department of Medicine, Seoul National University College of Medicine, Seoul 03080, Republic of Korea; 4Department of Family Medicine, Seoul National University Health Service Center, Seoul 08826, Republic of Korea; 5Department of Family Medicine, Seoul National University Hospital, Seoul 03080, Republic of Korea; 6Department of Family Medicine & Supportive Care Center, Samsung Medical Center, Sungkyunkwan University School of Medicine, Seoul 06351, Republic of Korea; 7Department of Clinical Research Design & Evaluation, Samsung Advanced Institute for Health Science & Technology (SAIHST), Sungkyunkwan University, Seoul 06355, Republic of Korea; 8Department of Digital Health, Samsung Advanced Institute for Health Science & Technology (SAIHST), Sungkyunkwan University, Seoul 06355, Republic of Korea; 9Department of Statistics and Actuarial Science, Soongsil University, Seoul 06978, Republic of Korea; 10Department of Biomedicine & Health Science, The Catholic University of Korea, Seoul 06591, Republic of Korea; 11Department of Family Medicine, Healthcare System Gangnam Center, Seoul National University Hospital, Seoul 06236, Republic of Korea; 12Department of Internal Medicine, Cardiovascular Center, Seoul National University Hospital, Seoul 03080, Republic of Korea

**Keywords:** alcohol consumption, change in drinking, heart failure, cohort

## Abstract

Background: The association between alcohol intake and newly developed heart failure remains unclear. We aimed to measure the change in alcohol intake between two timepoints to evaluate the association of alcohol consumption with incident heart failure using a population-based study in Korea. Methods: Using the Korean National Health Insurance database, participants who underwent two subsequent national health examinations in 2009 and 2011 were included. Participants were classified into four groups according to total alcohol intake (none: 0 g alcohol/day; light: <15 g alcohol/day; moderate: 15–30 g alcohol/day; and heavy: ≥30 g alcohol/day), and changes in alcohol consumption between the two health exams were grouped into the following five categories: abstainers, sustainers (those who maintained their first examination drinking level), increasers, reducers, and quitters. After adjustment for age, sex, smoking status, regular exercise, socioeconomic information, and comorbidities, the Charlson Comorbidity Index, systolic blood pressure, and laboratory results, a Cox proportional hazards model was used to find the risk of newly diagnosed heart failure (according to ICD-10 code I50 from claims for the first hospitalization) as the primary endpoint. A subgroup analysis among those with a third examination was conducted to reflect further changes in alcohol consumption. Results: Among 3,842,850 subjects, 106,611 (3.0%) were diagnosed with heart failure during the mean follow-up period of 6.3 years. Increasers to a light level of drinking had a lower HF risk compared with abstainers (aHR = 0.91, 95% CI: 0.89–0.94). Those who increased their alcohol intake to a heavy level had a higher HF risk (from light to heavy (aHR = 1.19, 95% CI: 1.12–1.26) and from a moderate to heavy level (aHR = 1.13, 95% CI: 1.07–1.19). Reducing alcohol from a heavy to moderate level was associated with lower HF risk (aHR = 0.90, 95% CI: 0.86–0.95). Conclusion: This study found that light and moderate sustainers had lower incident heart failure risk compared with abstainers. Increased alcohol consumption from light to moderate to heavy was associated with a higher incident heart failure risk.

## 1. Introduction

Heart failure (HF) carries substantial morbidity and mortality risks. In the general population, the prevalence ranges from 2% to over 10% in persons aged over 70 [1,2]. A common risk factor for HF is cardiovascular disease (CVD, e.g., coronary artery disease (CAD), hypertension, arrhythmia, or cardiomyopathy) [3,4,5]. Modifiable factors for CVD, such as obesity, smoking, and alcohol intake, are also thought to be linked to HF development [6,7].

Alcohol intake, a modifiable HF risk factor, could have different associations depending on the amount. Although long-term excessive drinking that causes structural and functional deterioration, including left ventricular hypertrophy [8] and left ventricular diastolic dysfunction [9], leads to alcoholic cardiomyopathy and, ultimately, HF, epidemiologic evidence for alcohol consumption indicates a potentially favorable effect from light-to-moderate drinking. A meta-analysis of eight longitudinal studies also reported that light-to-moderate alcohol drinking was inversely associated with HF risk [10]. Contrarily, the British Regional Heart Study found no evidence that light-to-moderate drinking is beneficial for HF prevention in healthy older men without a history of CVD [11]. The Framingham Heart Study reported that alcohol consumption was not associated with a higher HF risk even among heavy drinkers [12]. Therefore, the association between alcohol intake and HF remains unclear. In Korea, where alcohol consumption rates tend to exceed those reported elsewhere [13], associations between alcohol consumption, changes in consumption amounts, and HF risk have not been fully explored.

Indeed, most longitudinal studies that have explored the effect of alcohol intake on HF development used a one-time measure of alcohol consumption for their exposure group and made comparisons with lifetime abstainers or nondrinkers (summarized in Appendix A) [10,11,12,14,15,16,17,18,19,20,21,22,23,24]. No previous studies have evaluated the association between changes in alcohol consumption over time and HF risk. One way to elucidate the true association between alcohol intake and HF risk in an observational study is to explore the effects of changes in alcohol consumption by comparing current drinkers according to habit changes, including the initiation and cessation of drinking.

Therefore, we measured changes in alcohol intake at two timepoints to further evaluate the association between alcohol consumption and incident HF using a population-based study in Korea.

## 2. Materials and Methods

### 2.1. Data Source and Study Setting

This retrospective cohort study was based on the Korean National Health Insurance (KNHI) database. This cohort includes data on inpatient visits, outpatient visits, procedures, and prescription medications covered by the KNHI, a mandatory universal public health insurance system that covers the entire Korean population except for Medicaid beneficiaries in the lowest income bracket (3% of the population). The KNHI provides biennial national cardiovascular health screening for all beneficiaries aged 40 and above [25]. During biennial health screenings, KNHI study subjects respond to self-administered questionnaires for lifestyle factors (e.g., alcohol consumption, smoking, and physical activity) and medical and family history. The information on anthropometric measurements (blood pressure, body weight, and height), and laboratory tests (blood glucose, lipid profile, serum creatinine) are also collected.

The KNHI database contains a qualification database (e.g., age, sex, income, region, and type of eligibility), a claims database (e.g., general information on specifications, consultation statements, diagnosis statements defined by the International Classification of Diseases, 10th Revision (ICD-10), and prescription statements), a health checkup database, and death information. Our study was approved by the Institutional Review Board of Samsung Medical Center (File No. SMC 2019-02-059).

### 2.2. Study Population

A total of 4,961,817 participants who were aged 40 and above and who had undergone two subsequent national health examinations (2009, first exam, and 2011, second exam) were included from the KNHI database. Of these, participants who had been diagnosed previously with cancer (N = 144,259) or CVD (stroke, N = 307,183; myocardial infarction, N = 95,008; or HF, N = 116,918) before their second health exam were excluded. Additionally, to minimize possible reverse causality (e.g., sick quitter effect), participants who were diagnosed with any cancer (N = 43,767) or CVD (stroke, N = 33,637; myocardial infarction, N = 10,459; or HF, N = 3029) or who died (N = 4948) within one year after their second health exam (called one-year lag time) were excluded. In addition, patients with unavailable information on alcohol consumption and other covariate variables (N = 361,765) were excluded. Finally, a total of 3,842,850 subjects were included in the analyses (Figure 1).

### 2.3. Definition of Change in Alcohol-Intake Level

The average frequency of alcohol intake (how many times you usually drink per week) and the typical number of standard drinks consumed in a single session (how many glasses you drink per occasion) were obtained using self-reported questionnaires. Pure alcohol in grams was calculated based on the number of standard drinks, which is approximately 8 g for a typical volume of beer, wine, soju (a traditional Korean alcoholic drink), or whisky [26]. By multiplying the weekly frequency with the pure alcohol amount per occasion, the total amount of alcohol consumed per week was calculated.

According to total alcohol intake, participants were categorized into four groups: (1) none: 0 g alcohol/day; (2) light: <15 g alcohol/day; (3) moderate: 15–30 g alcohol/day; and (4) heavy: ≥30 g alcohol/day [27]. Consequently, 16 categories (4 × 4) were defined according to the 2009 and 2011 alcohol consumption statuses.

Additionally, changes in alcohol consumption between the two health exams were grouped into the following five categories: (1) abstainers, (2) sustainers (those who maintained their first-exam drinking level), (3) increasers (those who increased their drinking level), (4) quitters (those who stopped drinking), and (5) reducers (those who reduced their level of consumption but did not stop). This categorization was used to compare baseline characteristics with drinking behavior changes [28,29].

### 2.4. Covariates

To control confounders, covariates based on results from the second exam were included in the final analysis. Body mass index (BMI) was calculated as weight (kg) divided by height squared (m^2^). Smoking status was categorized into nonsmoker, former smoker, and current smoker. The average amount of daily cigarette smoking was also collected. Physical activity was defined as either >30 min of moderate physical activity at least five times/week or >20 min of strenuous physical activity at least three times/week during the past week. Comorbidities were defined by the medical claims according to the 10th edition of the ICD-10 codes (for hypertension, I10–I13 or I15; for diabetes, E11–E14; and for dyslipidemia, E78) and by prescriptions for relevant medications (e.g., antihypertensives). Insurance premium levels, which are determined using income levels by the Social Health Insurance System, were grouped into quartiles, and participants who received Medical Aid were included in the lowest quartile.

### 2.5. Definition of Heart Failure and Follow-Up

Newly diagnosed HF was the primary endpoint of this study. During follow-up, newly diagnosed HF was identified according to ICD-10 code I50 from claims for the first hospitalization. Participants were followed from the date of the second exam to the date of the HF diagnosis, the end of follow-up, or the end date of the study (31 December 2018), whichever came first.

### 2.6. Statistical Analyses

Categorical variables are presented as the number and percentage, and continuous variables are presented as the mean ± standard deviation (SD). The association between changes in alcohol consumption and incident HF was presented as a hazard ratio (HR) with a 95% confidence interval (CI) using the Cox proportional hazards model. Two types of reference groups were set to compare risk: (1) abstainers across all categories and (2) sustainers at the same level (e.g., mild–mild) within each category to determine risk according to both the level and change in level. Multivariable models adjusting for age, sex, BMI, smoking status, regular physical activity, area of residence, income, comorbidities (hypertension, diabetes mellitus, and dyslipidemia), systolic blood pressure, and laboratory findings (fasting glucose and total cholesterol) were established. We also performed a stratified analysis using possible effect modifiers (age, sex, and smoking habits) to identify any effect modifications in the association between alcohol intake and HF risk. Statistical analyses were performed using SAS version 9.4 (SAS Institute Inc., Cary, NC, USA), and a *p*-value < 0.05 was considered statistically significant.

## 3. Results

### 3.1. Baseline Characteristics

Clinical characteristics of the study population are presented in Table 1 according to the five groups based on changes in alcohol consumption (abstainers, quitters, reducers, sustainers, and increasers). At the first exam, 54.6%, 26.8%, 11.0%, and 7.6% of participants were nondrinkers or light, moderate, and heavy drinkers, respectively. Abstainers had a higher mean age and were more likely to be women (74.2%) and nonsmokers (84.6%). They had a higher prevalence of hypertension (33.4%) and dyslipidemia (25.7%) compared with other groups.

### 3.2. Risk of Heart Failure by Change in Alcohol Consumption: Comparisons with Abstainers

During the mean follow-up period of 6.3 ± 0.7 years, 106,611 individuals (3.0%) were diagnosed with HF. Compared with abstainers as a reference, those who sustained light-to-moderate drinking showed the lowest HF risk (adjusted HR (aHR) = 0.80, 95% CI: 0.79–0.82 in sustainers with a light intake level; aHR = 0.84, 95% CI: 0.81–0.87 in sustainers with a moderate intake level) (Appendix A).

### 3.3. Risk of Heart Failure by Change in Alcohol Consumption: Comparison with Sustainers

Subjects who initiated light alcohol consumption after their first exam showed a lower HF risk compared with abstainers (aHR = 0.91, 95% CI: 0.89–0.94), whereas HF risk among subjects who initiated drinking at moderate (aHR 0.97, 95% CI: 0.92–1.04) and heavy levels (aHR = 1.02, 95% CI: 0.95–1.10) was not significantly different from that of abstainers.

Using sustainers as the reference group, those who increased their consumption to heavy alcohol intake showed higher HF risk (from light to heavy intake (aHR = 1.19, 95% CI: 1.12–1.26) and from moderate to heavy intake (aHR = 1.13, 95% CI: 1.07–1.19)) (Figure 2 and Appendix A).

Reducing alcohol from heavy to moderate intake was associated with lower HF risk (aHR = 0.90, 95% CI: 0.86–0.95) compared with heavy drinking sustainers. In contrast, reducing alcohol from heavy to light intake (aHR = 1.05, 95% CI: 0.99–1.11) or from moderate to light intake (aHR = 1.03, 95% CI: 0.98–1.08) was associated with a slightly higher HF risk, although the results were only marginally significant.

Quitters had a higher HF risk relative to sustainers at each level: light (aHR = 1.19, 95% CI: 1.15–1.23), moderate (aHR = 1.37, 95% CI: 1.28–1.46), and heavy (aHR = 1.31, 95% CI: 1.22–1.41).

### 3.4. Stratified Analyses by Age, Sex, and Smoking Status

Figure 3, Figure 4 and Appendix A show results from stratified analyses in terms of effect modifiers (age at first exam, sex, and smoking habit). In both the age groups, initiation to light drinking was associated with lower HF risk (aHR = 0.91, 95% CI: 0.88–0.95 in younger ages, <65; aHR = 0.90, 95% CI: 0.86–0.94 in older ages, ≥65). Increased HF risk when increasing from light/moderate to heavy drinking was similar but slightly higher in older subjects: the aHR (95% CI) from light to heavy was 1.16 (1.08–1.26) for younger vs. 1.21 (1.09–1.33) for older; the aHR (95% CI) from moderate to heavy was 1.11 (1.04–1.19) for younger vs. 1.15 (1.03–1.27) for older (Figure 3 and Appendix A).

The initiation of light drinking was associated with lower HF risk in men (aHR = 0.88, 95% CI: 0.85–0.91) but this was not significant in women (aHR = 0.96, 95% CI: 0.92–1.01). Increased HF risk among increasers was more pronounced in women than men: aHR (95% CI) from light to moderate (1.04 (1.00–1.09) in men vs. 1.17 (1.03–1.33) in women); light to heavy (1.18 (1.11–1.26) in men vs. 1.37 (1.09–1.73) in women); and moderate to heavy (1.13 (1.06–1.19) in men] vs. 1.22 (0.91–1.65) in women). Additionally, higher HF risk among quitters was less prominent in women than men (Figure 4 and Appendix A). No notable differences were found in stratified analyses according to smoking status (Appendix A and Appendix A).

## 4. Discussion

To the best of our knowledge, this is the first study to assess HF risk based on changes in alcohol consumption measured at two timepoints. Our study confirmed that people with light-to-moderate alcohol consumption had the lowest HF risk, consistent with the results from previous studies. We also provided additional evidence that (1) the initiation of light alcohol intake is associated with lower HF risk than abstainers; (2) increasing to heavy drinking is associated with higher HF risk than sustaining light-to-moderate drinking; (3) reducing intake from heavy drinking is associated with lower HF risk than sustaining heavy drinking; and (4) quitting from any level was associated with higher HF risk than sustaining alcohol intake. Stratified analyses suggest that these effects were more prominent in older people and that women might be more susceptible to the harmful effects of heavy alcohol consumption.

Consistent with this, our findings about initiators and reducers to light and moderate intake supported previous evidence for a positive effect of light-to-moderate alcohol intake. Although the physiologic mechanisms supporting this association are unknown, some effects related to alcohol consumption can be suggested. First, the positive effect of alcohol may be mediated by improved CVD risk factors, which results in reduced HF risk. Improved insulin sensitivity [30] and higher HDL-cholesterol [31] have been suggested as underlying mechanisms as risk factors for CVD, as have decreased fibrinogen levels [32] and inflammatory markers [32,33]. Additionally, the intake of small amounts of alcohol increases atrial natriuretic peptide levels against HF progression [34]. However, the HF risk reduction with moderate alcohol consumption is explained partially by CAD risk reduction, which is a common antecedent of HF [14]. However, consistently reduced HF risk without preceding MI in light drinking does not entirely explain the favorable effect of alcohol on HF, suggesting that alcohol primarily affects HF development [12]. Direct links between moderate drinking and HF development require further study.

### 4.1. Harmful Effects of Heavy Drinking

This study indicates that sustainers with heavy drinking had slightly lower HF risk than abstainers but higher risk than light or moderate drinkers, suggesting that the protective effect of alcohol does not exist for heavy use. The Framingham Heart Study [12] and the Atherosclerosis Risk in Communities (ARIC) study [20] also showed nonsignificant lower HF risk in heavy drinkers. A study from the Kaiser Permanente Medical cohort reported a lower HF risk among people consuming fewer than one drink per day, whereas heavy alcohol consumption was associated with increased HF risk, particularly for non-CAD-related HF, possibly due to myocardial damage from heavy alcohol consumption [14].

Our study, which also investigated HF risk based on changes in alcohol consumption, suggested that heavy alcohol drinking might be harmful: increasers from light/moderate to heavy drinking had a higher HF risk than people who sustained their drinking levels, and subjects who initiated heavy drinking did not have the same protective effect as people who initiated light drinking.

Long-term heavy alcohol consumption can cause alcoholic cardiomyopathy, which is characterized by dilatation and wall thickness as well as increased left ventricle mass and a reduced left ventricular ejection fraction [35]. Moreover, the consumption of large amounts of alcohol can cause hypertension [7], which is an HF risk factor. Despite the finding that heavy drinkers might have better health and social profiles than abstainers [36], it is more likely that heavy alcohol consumption actually increases HF risk, and that previous studies with a single time measurement could not capture this due to confounding [12,20].

### 4.2. Quitting Alcohol Intake

Interestingly, quitting alcohol intake seemed to have a harmful effect. Our results also showed that quitters from all alcohol intake levels had higher HF risks relative to sustainers. This might be because of the “sick quitter” effect, where subjects who are diagnosed with some health conditions or experience symptoms begin abstaining from drinking, causing a reverse causality problem [37]. Slightly increased risks in reducers from heavy to light drinkers (aHR = 1.08, 95% CI: 1.00–1.17) compared with sustainers with heavy drinking can be understood in the same context, as they are likely to be have reduced their drinking rather than abstaining completely. Although we applied a one-year lag for our outcome assessment, it might not have been sufficient to account for bias from the observational study design.

### 4.3. Age Differences

The positive effect of initiating light alcohol drinking was found regardless of the age group. However, the initiation of moderate drinking did not seem to be favorable in older subjects. Our results were consistent with findings from longitudinal studies of older subjects that found no associations between light-to-moderate drinking and HF development [11,15] because the ability to metabolize alcohol declines with age, accompanied by decreased enzyme activity [38]. Changes in body water volume and slower alcohol elimination rates can cause higher blood alcohol concentrations at any intake level in elderly people compared with younger people [39]. Consequently, an increased serum ethanol level leads to a prolonged effect [40]. Prominent harmful effects of heavy drinking in older people compared with younger people in our study also indicate that elderly people suffer more from cardiac damage caused by alcohol toxicity than younger people do.

### 4.4. Gender Differences

The pronounced risk among female increasers in our study suggests that women are more susceptible to the toxic cardiac effects of alcohol. Women are thought to have a stronger association between alcohol consumption and HF than men [12,18,20]. Compared with men, women had worse left and right ventricular function for a given degree of alcohol consumption [20]. Biological properties explain that women develop alcohol-associated cardiomyopathy at lower lifetime alcohol doses compared with men [41] and have a much greater risk of heart-disease-related death [42]. Women might be susceptible to alcohol-related impacts reflecting higher blood alcohol concentrations for a given intake amount, which can be explained by their higher proportion of body fat; thus, women absorb and metabolize alcohol differently than men [43].

### 4.5. Limitations

This study has several limitations. First, this was an observational study, and alcohol consumption was self-reported in a questionnaire administered by an interviewer. Participants might have underreported their consumption levels. Second, as we defined incident HF based on the ICD-10 code, we could not classify HF cases based on underlying etiologies or supply clinical information such as cardiac function or structure due to a lack of information. Third, we did not consider alcohol drinking patterns, including binge drinking or beverage types, although drinking patterns might be important for HF risk.

## 5. Conclusions

This study found that sustainers at light-to-moderate levels had a lower incident HF risk than abstainers. In contrast, increased alcohol consumption from light/moderate to heavy consumption was associated with higher HF risk compared with light-to-moderate sustainers. The deleterious associations of heavy alcohol consumption with HF were more prominent in elderly people and women. However, it should be noted that alcohol usage should not be promoted as an HF prevention strategy given the other serious health risks related to drinking.

## Figures and Tables

**Figure 1 ijerph-19-16265-f001:**
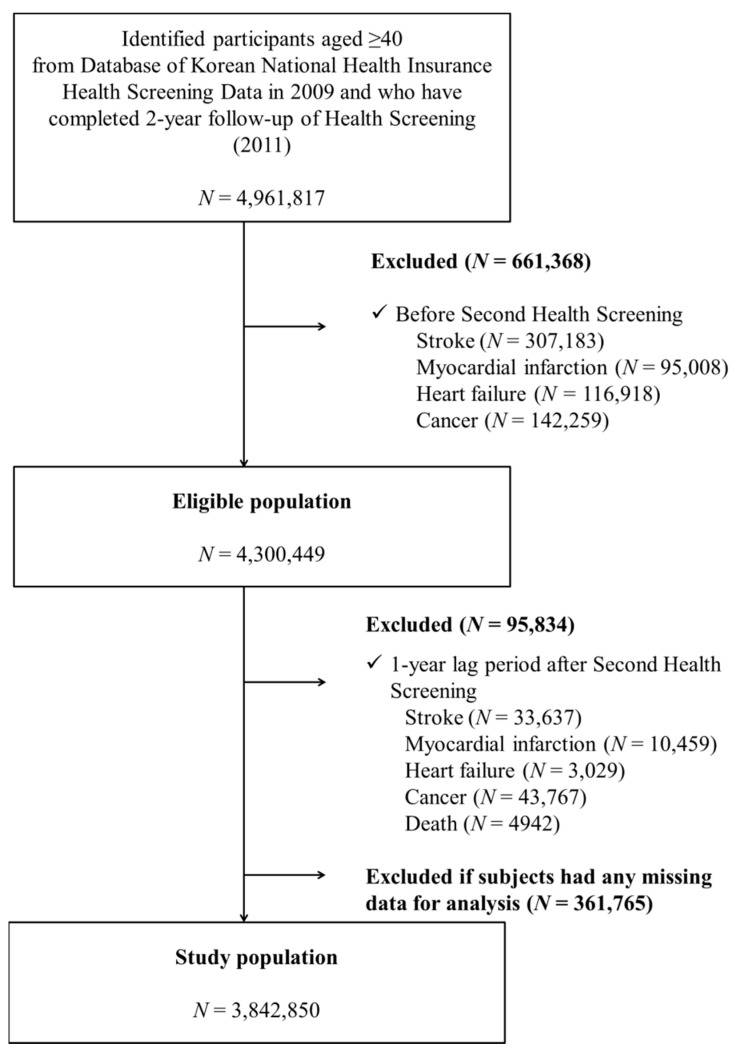
Study design.

**Figure 2 ijerph-19-16265-f002:**
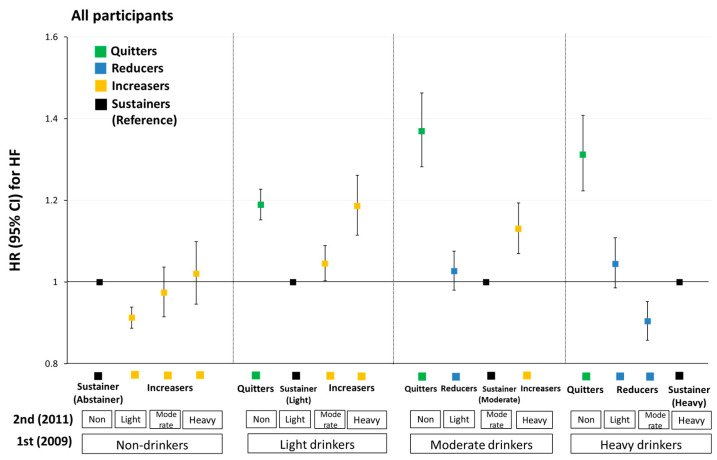
Hazard ratios (HRs) and 95% confidence intervals (CIs) for the association between change in alcohol consumption amount and risk of congestive heart failure. All HRs were adjusted for age, sex, BMI, smoking status, physical activity, area of residence, income, hypertension, diabetes mellitus, dyslipidemia, systolic blood pressure, fasting glucose level, total cholesterol level, and serum creatinine level.

**Figure 3 ijerph-19-16265-f003:**
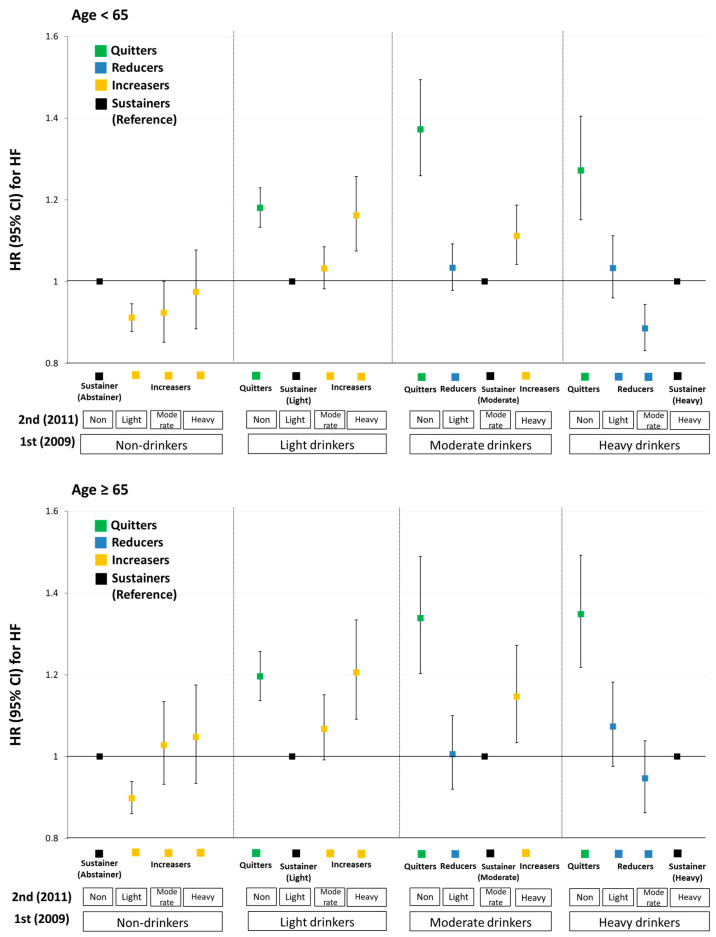
Changes in alcohol consumption amount and risk of congestive heart failure by age. All HRs were adjusted for age, sex, BMI, smoking status, physical activity, area of residence, income, hypertension, diabetes mellitus, dyslipidemia, systolic blood pressure, fasting glucose level, total cholesterol level, and serum creatinine level. HR, hazard ratio; CI, confidence interval.

**Figure 4 ijerph-19-16265-f004:**
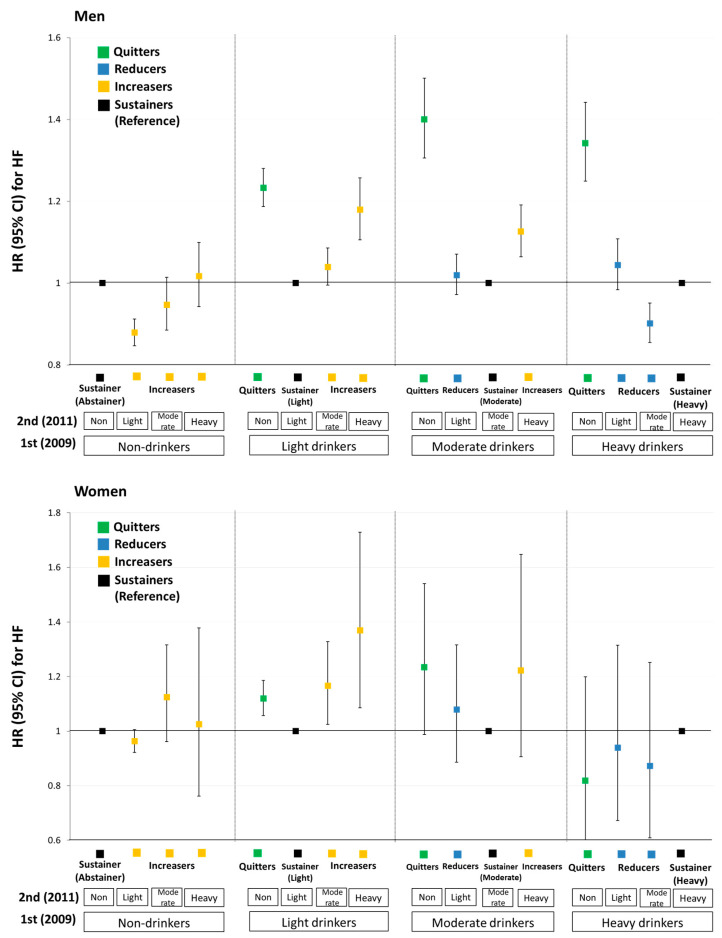
Changes in alcohol consumption amount and risk of congestive heart failure by sex. All HRs were adjusted for age, sex, BMI, smoking status, physical activity, area of residence, income, hypertension, diabetes mellitus, dyslipidemia, systolic blood pressure, fasting glucose level, total cholesterol level, and serum creatinine level. HR, hazard ratio; CI, confidence interval.

**Table 1 ijerph-19-16265-t001:** Baseline characteristics of the study population.

	Change in Alcohol Consumption on Subsequent Two Health Examinations					
Variables	Abstainers	Quitters	Reducers	Sustainers	Increasers	*p* Value
	(N = 1,801,711)	(N = 305,487)	(N = 279,756)	(N = 921,802)	(N = 534,094)	
Mean age, years (mean ± SD)	57.1 ± 9.9	54.7 ± 9.51	52.7 ± 8.5	52.1 ± 8.4	53.0 ± 8.8	<0.001
Men (N, %)	465,296 (25.8)	168,267 (55.1)	254,748 (91.1)	731,139 (79.3)	381,938 (71.5)	<0.001
Alcohol consumption in 2009 ^†^ (N, %)						
None	1,801,711 (100.0)	NA	NA	NA	293,439 (54.9)	<0.001
Light (<15 g/day)	NA	248,072 (81.2)	NA	616,935 (66.9)	166,672 (31.2)	
Moderate (15–29 g/day)	NA	35,445 (11.6)	148,992 (53.3)	166,252 (18.0)	73,983 (13.9)	
Heavy (≥30 g/day)	NA	21,970 (7.2)	130,764 (46.7)	138,615 (15.1)	NA	
Alcohol frequency in 2009 ^†^ (N, %)						
None	1,801,711 (100.0)	NA	NA	NA	293,439 (54.9)	<0.001
1–2 days/week	NA	252,159 (82.5)	107,267 (38.3)	663,079 (71.9)	170,129 (31.9)	
3–4 days/week	NA	33,549 (11.0)	115,054(41.1)	177,049 (19.2)	58,549 (11.0)	
5–7 days/week	NA	19,779 (6.5)	57,435 (20.5)	81,674 (8.9)	11,977 (2.2)	
Alcohol consumption amount per occasion in 2009 ^†^ (N, %)						
None	1,801,711 (100.0)	NA	NA	NA	293,439 (54.9)	<0.001
1–2 drinks	NA	108,462 (35.5)	2265 (0.8)	140,926 (15.3)	23,152 (4.3)	
3–4 drinks	NA	92,875 (30.4)	15,593 (5.6)	254,419 (27.6)	59,146 (11.1)	
5–7 drinks	NA	69,679 (22.8)	119,126 (42.6)	313,865 (34.1)	104,709 (19.6)	
≥8 drinks	NA	34,459 (11.3)	142,772 (51.0)	212,557 (23.0)	53,635 (10.1)	
Smoking status (N, %)						
None	1,523,503 (84.6)	214,390 (70.2)	72,942 (26.1)	345,310 (37.5)	233,659 (43.7)	<0.001
Former smoker	143,938 (8.0)	46,060 (15.1)	88,245 (31.5)	272,373 (29.5)	136,563 (25.6)	
Current, <10 cigarettes/day	16,888 (0.9)	6355 (2.0)	12,424 (4.4)	32,551 (3.5)	17,042 (3.2)	
Current, 10–19 cigarettes/day	49,823 (2.8)	17,405 (5.7)	47,488 (17.0)	122,194 (13.3)	60,155 (11.3)	
Current, ≥20 cigarettes/day	67,559 (3.7)	21,277 (7.0)	58,657 (21.0)	149,374 (16.2)	86,675 (16.2)	
Physical activity (N, %)						<0.001
None	1,435,610 (79.7)	242,945 (79.5)	210,334 (75.2)	695,520 (75.4)	405,783 (76.0)	
Moderate or vigorous intensity	271,384 (15.1)	45,502 (14.9)	52,538 (18.8)	172,323 (18.7)	95,913 (18.0)	
Moderate and vigorous intensity	94,717 (5.3)	17,040 (5.6)	16,884 (6.0)	53,959 (5.9)	32,398 (6.0)	
Anthropometrics (mean ± SD)						
Body mass index, kg/m^2^	23.8 ± 3.1	23.9 ± 2.9	24.3 ± 2.9	24.0 ± 2.8	24.1 ± 2.9	<0.001
WC, cm	79.4 ± 8.5	81.0 ±8.5	84.0 ± 7.6	82.5 ± 8.0	82.3 ±8.3	<0.001
SBP, mmHg	122.8 ± 15.2	122.9 ± 14.7	126.4 ± 14.2	124.5 ± 14.3	124.6 ± 14.7	<0.001
DBP, mmHg	75.7 ± 9.8	76.5 ± 9.8	79.4 ± 9.8	78.2 ± 9.8	78.1 ± 10.0	<0.001
Comorbidity (N, %)						
Hypertension	599,286 (33.4)	97,494 (31.9)	102,788 (36.7)	292,549 (31.7)	175,562 (32.9)	<0.001
Diabetes mellitus	192,576 (10.7)	33,787 (11.1)	36,364 (13.0)	95,973 (10.4)	60,210 (11.3)	<0.001
Dyslipidemia	462,493 (25.7)	67,774 (22.2)	58,547 (20.9)	181,123 (19.7)	108,434 (20.3)	<0.001
CKD	119,568 (6.6)	15,407 (5.0)	10,775 (3.9)	36,648 (4.0)	22,126 (4.1)	<0.001
Laboratory findings (mean ± SD)						
Glucose, mg/dL	97.9 ± 22.2	99.3 ± 23.7	102.9 ± 25.7	100.6 ± 23.7	101.0 ± 24.9	<0.001
Total cholesterol, mg/dL	199.9 ± 36.9	197.9 ± 36.5	197.8 ± 35.7	198.0 ± 35.1	198.2 ±35.6	<0.001
HDL, mg/dL	54.8 ± 16.6	54.5 ± 17.4	54.8 ± 17.4	55.3 ± 17.4	55.9 ± 18.3	<0.001
LDL, mg/dL	120.7 ± 33.8	117.9 ± 33.8	112.4 ± 34.7	114.7± 33.8	114.6 ± 34.0	<0.001
GFR, mL/min/1.73 m^2^	86.9 ± 31.0	87.9 ± 32.4	89.0 ± 38.1	88.3 ± 37.5	88.7 ± 35.2	<0.001
Urban residency, (N, %)	798,440 (44.3)	138,208 (45.2)	127,960 (45.7)	438,405 (47.6)	242,026 (45.3)	<0.001
Income, (N, %)						
Q1 (lowest)	472,769 (26.2)	74,015 (24.2)	50,931 (18.2)	172,525 (18.7)	114,141 (21.4)	<0.001
Q2	341,643 (19.0)	58,962 (19.3)	46,497 (16.6)	151,333 (16.4)	94,576 (17.7)	
Q3	411,182 (22.8)	71,332 (23.4)	70,834 (25.3)	217,490 (23.6)	126,751 (23.7)	
Q4 (highest)	576,117 (32.0)	101,178 (33.1)	111,494 (39.9)	380,454 (41.3)	198,626 (37.2)	

Data are presented as number (%) or mean ± standard deviation. BP, blood pressure; SBP, systolic blood pressure; DBP, diastolic blood pressure; GFR, glomerular filtration rate; CKD, chronic kidney disease; NA, nonapplicable; SD, standard deviation; Q, quartile; WC, waist circumference; HDL, high-density lipoprotein; LDL, low-density lipoprotein. ^†^ Alcohol consumption data were obtained via self-questionnaire at first examination in 2009.

## Data Availability

The researchers should submit a study proposal for acquiring approval from each institutional review board, which is also reviewed by the NHIS review committee to access the database. The datasets and statistical analysis used for the current study can be repeated only by a reanalysis at the data center upon reasonable request.

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
