# Peer review of "Changes in Alcohol Consumption and Risk of Heart Failure: A Nationwide Population-Based Study in Korea"

_ijerph, 2022, doi:10.3390/ijerph192316265_

Round 1
Reviewer 1 Report
The manuscript entitled" Changes in Alcohol Consumption and Risk of Heart Failure: A Nationwide Population-Based Study in Korea" is an interesting original paper investigating the association between changes of alcohol intake (during a 2 year period) and risk of HF using survival models. The results are interesting but often the scientific soundness and clarity is missing
Please see below comments to the manuscript.
Abstract
the classification made for alcohol changes is a bit confusing. it is not clear to me which is the reference category (is it always the same for all the group). The categorization of alcohol consumption in four categories is compared to the changes at the second examination? Also, the terminology used in the results section of the abstract such as light drinking after the first exam does not correspond to any of the classification mentioned earlier. Are those the increasers? Please be consistent. You might also want to consider using the term abstainer instead of non. I also think that the term "sustained" could be replace it with a more appropriate terminology.
Please insert also how HF was diagnosed.
It is missing the statistical method used. Is that multi-adjsuted cox proportional hazard models? Adjusted for which covariates? How were the changes in alcohol modelled?
Main text:
Introduction:
maybe it could be relevant to add information regarding alcohol intake habits in Korean population (and/or Asiatic vs other Americans or Europeans). Also, it could be interesting to have some short information how the Asian heart failure association (if exist or Asiatic CVD association) relate to alcohol intake. Are there any study conducted in asiatic on alcohol intake and HF? If not, please clarify it. The Supplementary Table 1 is very informative but could the authors summarize those details in the introduction?
Method section:
Covariates: please explain how these covariates were selected. For causal inference purposes, consider these covariates confounders?
Statistical analyses:
the authors state that two reference categories were considered for this study. Please expand it and give more information. Please explain the reason of doing it. Was one of the method added as sensitivity analyses.
Also expand the reason why stratified analyses were performed.
Results:
The figures presenting the associations between changes of alcohol and incidence of HF need some clarifications. The notes below the Fig reporting the amount of alcohol at baseline and at the first visit are a bit confusing. Overall they are very nice but I think they are not easy to read and interpret (as the table 2 in the SM).
Discussion/Conclusion
I would be very careful in discussing the beneficial effect of starting to drink alcohol. As you may know even the cvd guidelines regarding alcohol intake have been change and the one from the American heart Association suggest to not start to drink alcohol.
Author Response
November 14, 2022
Editor-in-Chief
International Journal of Environmental Research and Public Health
Thank you very much for your consideration of our manuscript, entitled “Changes in Alcohol Consumption and Risk of Heart Failure: A Nationwide Population-Based Study in Korea.” We also appreciate the constructive suggestions and comments we received from the editor and reviewers. We have addressed the reviewers’ comments point by point, and the revised sections are marked with tracking and highlights in the manuscript.
We hope that you and the reviewers find the revisions satisfactory.
Sincerely,
Dong Wook Shin, MD, DrPH, MBA
Department of Family Medicine/Supportive Care Center, Samsung Medical Center
Department of Digital Health, SAIHST, Sungkyunkwan University
81 Irwon-Ro, Gangnam-gu, Seoul 06351, Republic of Korea
Tel: 82-2-3410-5252; Fax: 82-2-3410-0388; E-mail:dwshin.md@gmail.com
Su-Min Jeong, MD
Department of Family Medicine & Supportive Care Center, Samsung Medical Center, Sungkyunkwan University School of Medicine, Seoul, Republic of Korea
81 Irwon-Ro, Gangnam-gu, Seoul 06351, Republic of Korea
Tel: 82-2-3410-5252; fax: 82-2-3410-0388; E-mail: smjeong.fm@gmail.com
Reviewer 1
The manuscript entitled" Changes in Alcohol Consumption and Risk of Heart Failure: A Nationwide Population-Based Study in Korea" is an interesting original paper investigating the association between changes of alcohol intake (during a 2 year period) and risk of HF using survival models. The results are interesting but often the scientific soundness and clarity is missing.
Please see below comments to the manuscript.
Comment #1. (Abstract)
1) the classification made for alcohol changes is a bit confusing. it is not clear to me which is the reference category (is it always the same for all the group). The categorization of alcohol consumption in four categories is compared to the changes at the second examination?
[Response] We added the reference level of alcohol intake in the Abstract section. Per the reviewer’s comment, the four categories of alcohol consumption (non-drinker, light, moderate, and heavy drinkers) were defined at two times (in the first examination in 2009 and the second one in 2011). After that, five categories were defined according to the changes in alcohol consumption between the two health examinations (non-drinkers, sustainers, increasers, reducers, and quitters).
“Participants were classified into four groups according to total alcohol intake (none: 0g alcohol/day, light: <15g alcohol/day, moderate: 15–30g alcohol/day, and heavy: ≥30g alcohol/day), and changes in alcohol consumption between the two health exams were grouped into the following five categories: abstainers, sustainers who maintained their first examination drinking level, increasers, reducers, and quitters.”
2) Also, the terminology used in the results section of the abstract such as light drinking after the first exam does not correspond to any of the classification mentioned earlier. Are those the increasers? Please be consistent.
[Response] Light drinkers at the second exam who were not drinkers at the first exam were classified as increasers based on the changes of alcohol consumption. In the results section in the Abstract, we clearly revised the phrase.
“Increasers to a light level drinking had a lower HF risk compared with abstainers (aHR=0.91, 95% CI: 0.89–0.94).”
3) You might also want to consider using the term abstainer instead of non. I also think that the term "sustained" could be replace it with a more appropriate terminology.
[Response] We replaced “sustained non-drinkers” to “abstainers” throughout the manuscript.
Comment #2. (Abstract) Please insert also how HF was diagnosed.
[Response] In this study, we identified patients who were hospitalized for HF using a claims database. This criterion has been widely adopted in epidemiological studies [Yoo et al JACC Heart Failure 2022].
We also added the definition on HF diagnosis in the Abstract.
“After adjustment for age, sex, smoking status, regular exercise, socioeconomic information, and comorbidities, Charlson Comorbidity Index, systolic blood pressure, and laboratory results, Cox proportional hazards model was used to find the risk of newly diagnosed heart failure (according to ICD-10 code I50 from claims for the first hospitalization) as the primary endpoint. Subgroup analysis among those with the third examination was conducted to reflect further change in alcohol consumption.”
References
1) Yoo J, Jeong S, Yeo Y, et al. Smoking Cessation Reduces the Risk of Heart Failure. J Am Coll Cardiol HF. Online ahead..https://doi.org/10.1016/j.jchf.2022.07.006
Comment #3. (Abstract) It is missing the statistical method used. Is that multi-adjsuted cox proportional hazard models? Adjusted for which covariates? How were the changes in alcohol modelled?
[Response] In the Abstract, we added statistical methods and covariates that were used in our study.
“After adjustment for age, sex, smoking status, regular exercise, socioeconomic information, and comorbidities, Charlson Comorbidity Index, systolic blood pressure, and laboratory results, Cox proportional hazards model was used to find the risk of newly diagnosed heart failure (according to ICD-10 code I50 from claims for the first hospitalization) as the primary endpoint. Subgroup analysis among those with the third examination was conducted to reflect further change in alcohol consumption.”
In addition, we modified our definition of changes in alcohol as follows.
The same definition was used in our previous studies [Jeong SM, et al Stroke 2022; Yoo JE, et al JAMA Netw Open. 2022]. We describe this in the Methods section and cited these studies in the manuscript (Definition of change in alcohol intake level).
“Additionally, changes in alcohol consumption between the two health exams were grouped into the following five categories: 1) abstainers, 2) sustainers (those who maintained their first-exam drinking level), 3) increasers (those who increased their drinking level), 4) quitters (those who stopped drinking), and 5) reducers (those who reduced their level of consumption but did not stop). This categorization was used to compare baseline characteristics by behavior change.
References
1) Jeong SM, Lee HR, Han K, Jeon KH, Kim D, Yoo JE, Cho MH, Chun S, Lee SP, Nam KW, Shin DW. Association of Change in Alcohol Consumption With Risk of Ischemic Stroke. Stroke. 2022 Aug;53(8):2488-2496. doi: 10.1161/STROKEAHA.121.037590.
2) Yoo JE, Han K, Shin DW, Kim D, Kim BS, Chun S, Jeon KH, Jung W, Park J, Park JH, Choi KS, Kim JS. Association Between Changes in Alcohol Consumption and Cancer Risk. JAMA Netw Open. 2022 Aug 1;5(8):e2228544.
Comment #4. (Main text: Introduction) maybe it could be relevant to add information regarding alcohol intake habits in Korean population (and/or Asiatic vs other Americans or Europeans). Also, it could be interesting to have some short information how the Asian heart failure association (if exist or Asiatic CVD association) relate to alcohol intake. Are there any study conducted in asiatic on alcohol intake and HF? If not, please clarify it. The Supplementary Table 1 is very informative but could the authors summarize those details in the introduction?
[Response] Alcohol consumption in Korea has been reported to be higher. According to the Global Status Report on Alcohol and Health 2018 [World Health Organization], annual alcohol consumption per capita in Korea is 21.7 L, the 15th highest rate worldwide. Since the early 2010s, an increasing trend in alcohol drinking rates was reported in the representative Korean health survey. In the Korea National Health and Nutrition Examination Survey (KNHANES), monthly alcohol drinking of adults was reported to be 59.0%, and the prevalence of high-risk alcohol drinking was 18.0% [Ministry of Health and Welfare; 2018; Ryu SY et al, J Prev Med Public Health. 2013]. However, oriental cultural circumstances usually show toleration to alcohol drinking and it is common to not recognize problems with alcohol use (dependence or disorder) in clinical setting [Hong JW et al. PLoS One 2017].
Despite a lower alcohol consumption rate in women than men in Asian countries (most abstainers were women (78.0%)), harmful use and alcohol use disorder are also prominent in women [Lee S et al. Korean J Fam Med. 2019]. A study by Choe SA et al of 34,478 subjects reported that the proportion of harmful alcohol use in men decreased, but did not do so in women [Choe SA et al. JKMS 2018]. The prejudice from that women usually do not drink can act as a cultural barrier that overlooks the establishment of health strategies against alcohol drinking in Korea [Kim W et al. Subst Use Misuse 2008; Ham BJ et al. Soc Psychiatry Psychiatr Epidemiol 2005].
Until now, unfortunately, the association between alcohol drinking and HF was not explored in Korea. The health effect of alcohol consumption has been reported for various health outcomes (all-cause mortality, cancers, type 2 diabetes mellitus, hyperlipidemia, or alcohol use problem and injury). Recently, studies reported that alcohol consumption is associated with CVD [Jeong SM, et al Stroke 2022] and cancer [Yoo JE, et al JAMA Netw Open 2022], but the effect of alcohol drinking was relatively diverse according to both intake level and target diseases. For this reason, we explored the association between alcohol consumption and incident HF using a population-based study in Korea regarding possible effect modifiers (including age, and sex).
We briefly mentioned the lack of evidence on the association between alcohol intake and HF risk in Korea in the Introduction Section.
“In Korea, where alcohol consumption rates tend to exceed those reported elsewhere [13], the association between alcohol consumption change in consumption amounts, and HF risk has not been fully explored.”
References
1) Global status report on alcohol and health 2018. Geneva: World Health Organization; 2018. Licence: CC BY-NC-SA 3.0 IGO
2) Ministry of Health and Welfare; Korea Centers for Disease Control and Prevention. Korea health statistics 2017: Korea National Health and Nutrition Examination Survey (KNHANES IV-3). Seoul: Ministry of Health and Welfare; 2018, p.22-23.
3) Ryu SY, Crespi CM, Maxwell AE. Drinking Patterns Among Korean Adults: Results of the
2009 Korean Community Health Survey. J Prev Med Public Health. 2013 Jul; 46(4): 183–191.
4) Hong JW, Noh JH, Kim DJ. The prevalence of and factors associated with high-risk alcohol consumption in Korean adults: the 2009–2011 Korea National Health and Nutrition Examination Survey. PLoS One 2017;12(4):e0175299.
5) Lee S, Kim JS, Oh MK, Ching TH, Kim J. Korean Alcohol Guidelines for Moderate Drinking Based on Facial Flushing. Korean J Fam Med. 2019; 40(4): 204–211.
6) Choe SA, Yoo S, JeKarl J, Kim KK. Recent Trend and Associated Factors of Harmful Alcohol Use Based on Age and Gender in Korea. J Korean Med Sci. 2018 Jan 22;33(4):e23.
7) Kim W, Kim S. Women's alcohol use and alcoholism in Korea. Subst Use Misuse 2008;43(8-9):1078-87.
8) Hahm BJ, Cho MJ. Prevalence of alcohol use disorder in a South Korean community: changes in the pattern of prevalence over the past 15 years. Soc Psychiatry Psychiatr Epidemiol 2005;40(2):114-119.
9) Jeong SM, Lee HR, Han K, Jeon KH, Kim D, Yoo JE, Cho MH, Chun S, Lee SP, Nam KW, Shin DW. Association of Change in Alcohol Consumption With Risk of Ischemic Stroke. Stroke. 2022 Aug;53(8):2488-2496. doi: 10.1161/STROKEAHA.121.037590.
10) Yoo JE, Han K, Shin DW, Kim D, Kim BS, Chun S, Jeon KH, Jung W, Park J, Park JH, Choi KS, Kim JS. Association Between Changes in Alcohol Consumption and Cancer Risk. JAMA Netw Open. 2022 Aug 1;5(8):e2228544.
Comment #5. (Method section-Covariates) please explain how these covariates were selected. For causal inference purposes, consider these covariates confounders?
[Response] Per the reviewer’s comment, we selected covariates of common risk factors for HF [McMurray JJ. Lancet 2005]. Possible predisposed diseases (hypertension, diabetes mellitus, and dyslipidemia) and their modifiable risk factors of obesity, smoking, and physical activity were included as confounders ) [McManus DD. JACC 2016; Larsson SC et al. JACC 2014; Ruidavets J-B et al. BMJ 2010; Briasoulis A et al. J Clin Hypertens 2012].
We briefly mentioned the description in the Methods Section (a brief purpose in Covariates and detailed in Statistical analysis).
“To control for confounders, covariates based on results from the second exam were included in the final analysis.”
“Multivariable models adjusting for age, sex, BMI, smoking status, regular physical activity, area of residence, income, comorbidities (hypertension, diabetes mellitus, and dyslipidemia), systolic blood pressure, and laboratory findings (fasting glucose and total cholesterol) were established.”
References
1) McMurray JJ, Pfeffer MA. Heart failure. Lancet. 2005, 365, 1877-1889.
2) McManus DD, Yin X, Gladstone R, Vittinghoff E, Vasan RS, Larson MG, Benjamin EJ, Marcus GM. Alcohol Consumption, Left Atrial Diameter, and Atrial Fibrillation. J Am Heart Assoc. 2016, 5.
3) Larsson SC, Drca N, Wolk A. Alcohol consumption and risk of atrial fibrillation: a prospective study and dose-response meta-analysis. J Am Coll Cardiol. 2014, 64, 281-289.
4) Ruidavets J-B, Ducimetière P, Evans A, Montaye M, Haas B, Bingham A, Yarnell J, Amouyel P, Arveiler D, Kee F, et al. Pat-terns of alcohol consumption and ischaemic heart disease in culturally divergent countries: the Prospective Epidemiologi-cal Study of Myocardial Infarction (PRIME). BMJ. 2010, 341, c6077.
5) Briasoulis A, Agarwal V, Messerli FH. Alcohol consumption and the risk of hypertension in men and women: a systematic review and meta-analysis. J Clin Hypertens (Greenwich). 2012, 14, 792-798.
Comment #6. (Statistical analyses)
1) the authors state that two reference categories were considered for this study. Please expand it and give more information. Please explain the reason of doing it. Was one of the method added as sensitivity analyses.
[Response] Considering the likelihood of changes in alcohol consumption habits over time, it is important to understand how changes are associated with risk of HF. Because the effect of alcohol intake on HF might depend on the amount of intake, the association between alcohol intake and HF remains unclear despite findings that light-to-moderate drinking would be favorable to HF. To determine the true effect of alcohol intake on HF, a two-step comparison was performed regarding 1) alcohol intake level at the first exam and 2) change in alcohol intake and their level.
One method of comparison is using lifetime abstainers or non-drinkers as a reference (usually used in a study of one-time measurement of alcohol intake). The risk of HF according to alcohol intake level at the first examination compared to abstainers (non-drinkers) showed that light to moderate level sustainers had lower risk of HF (HR 0.84 for light, 0.84 for moderate level sustainers).
Second, the main result showed a similar risk pattern regarding both change and level of change in alcohol intake stratified by the reference of sustainers in subsequent exams. We conclude that sustainers at light-to-moderate levels had lower incident HF risk than abstainers (non-drinkers) and that increase in alcohol consumption from light/moderate to heavy level was associated with higher HF risk compared with light-to-moderate sustainers.
We explained the use of these two techniques in the Methods Section (Statistical Analysis).
“Two types of reference groups were set to compare risk: 1) abstainers across all categories and 2) sustainers at the same level (e.g., mild-mild) within each category to determine risk according to both level and change in level.”
2) Also expand the reason why stratified analyses were performed.
[Response] We performed stratified analysis by a possible effect modifiers (age at first exam, sex, and smoking habits) to find the interactions between alcohol intake and HF risk. We found that the deleterious associations of heavy alcohol consumption with HF were more prominent in elderly people and women.
We clearly described the purpose of stratified analysis in the Methods Section (Statistical analysis)
“We also performed stratified analysis using possible effect modifiers (age, sex, and smoking habits) to identify any effect modification in the association between alcohol intake and HF risk.”
Comment #7. (Results) The figures presenting the associations between changes of alcohol and incidence of HF need some clarifications. The notes below the Fig reporting the amount of alcohol at baseline and at the first visit are a bit confusing. Overall they are very nice but I think they are not easy to read and interpret (as the table 2 in the SM).
[Response] We clarified the description notes in Figures by including both the five categories by the change of alcohol intake and the amount of alcohol consumption at the two-time measurements.
(Figure 2)
(Figure 3)
(Figure 4)
Comment #8. (Discussion/Conclusion) I would be very careful in discussing the beneficial effect of starting to drink alcohol. As you may know even the cvd guidelines regarding alcohol intake have been change and the one from the American heart Association suggest to not start to drink alcohol.
[Response] For careful interpretation of the results regarding the overall health effect beyond the HF risk, we added a clear description in the Conclusion Section.
“However, it should be noted that alcohol usage should not be promoted as HF prevention strategies, given that this choice would be associated with other serious health risks.”
“However, alcohol usage should not be promoted as an HF prevention strategy, given the other serious health risks related to drinking.”
Please see the attachment.

Reviewer 2 Report
Dear authors,
The manuscript by Yohwa Yeo et al. is an interesting study aiming to investigate the potential effect of changes in alcohol consumption and the risk of heart failure. The manuscript is well-written and flows easily. Minor issues with typos and optimized syntax exist and proofreading potentially by an English native speaker is suggested.
Apart from this, after using an anti-plagiarism tool, I have found that there are literally copied paragraphs from these articles such as:
8% from:
Jeong SM, Lee HR, Han K, et al. Association of Change in Alcohol Consumption With Risk of Ischemic Stroke. Stroke. 2022;53(8):2488-2496. doi:10.1161/STROKEAHA.121.037590
4% from:
Gonçalves A, Claggett B, Jhund PS, et al. Alcohol consumption and risk of heart failure: the Atherosclerosis Risk in Communities Study. Eur Heart J. 2015;36(15):939-945. doi:10.1093/eurheartj/ehu514
I attach the plagiarism report that shows this issue.
I have some concerns about some issues:
Which self-reported questionnaires were used to calculate the frequency of alcohol intake?
In section 2.4: covariates
I think tobacco consumption is briefly described and should have an accurate description.
Although some other covariates are included in the analysis, they are not mentioned in the article, only in Table 1 such as smoking status, physical activity, BMI … I think it would be interesting to correlate them.
References:
Please check the journal´s recommendations, all the references don´t follow the ACS style guide.

Author Response
November 14, 2022
Editor-in-Chief
International Journal of Environmental Research and Public Health
Thank you very much for your consideration of our manuscript, entitled “Changes in Alcohol Consumption and Risk of Heart Failure: A Nationwide Population-Based Study in Korea.” We also appreciate the constructive suggestions and comments we received from the editor and reviewers. We have addressed the reviewers’ comments point by point, and the revised sections are marked with tracking and highlights in the manuscript.
We hope that you and the reviewers find the revisions satisfactory.
Sincerely,
Dong Wook Shin, MD, DrPH, MBA
Department of Family Medicine/Supportive Care Center, Samsung Medical Center
Department of Digital Health, SAIHST, Sungkyunkwan University
81 Irwon-Ro, Gangnam-gu, Seoul 06351, Republic of Korea
Tel: 82-2-3410-5252; Fax: 82-2-3410-0388; E-mail:dwshin.md@gmail.com
Su-Min Jeong, MD
Department of Family Medicine, Seoul National University Health Service Center, Seoul, Republic of Korea.
101 Daehak-Ro, Jongro-gu, Seoul 03080, Republic of Korea
Tel: 82-2-3410-5252; fax: 82-2-3410-0388; E-mail: smjeong.fm@gmail.com
Reviewer 2
Dear authors,
The manuscript by Yohwan Yeo et al. is an interesting study aiming to investigate the potential effect of changes in alcohol consumption and the risk of heart failure. The manuscript is well-written and flows easily.
Comment #1.
1) Minor issues with typos and optimized syntax exist and proofreading potentially by an English native speaker is suggested.
[Response] We revised the manuscript to correct language errors based on the reviewer’s comment. We also had the manuscript revised by a commercial English-language editing service (eWorld Editing, Eugene, OR, USA).
2) Apart from this, after using an anti-plagiarism tool, I have found that there are literally copied paragraphs from these articles such as:
8% from: Jeong SM, Lee HR, Han K, et al. Association of Change in Alcohol Consumption With Risk of Ischemic Stroke. Stroke. 2022;53(8):2488-2496. doi:10.1161/STROKEAHA.121.037590
4% from: Gonçalves A, Claggett B, Jhund PS, et al. Alcohol consumption and risk of heart failure: the Atherosclerosis Risk in Communities Study.Eur Heart J. 2015;36(15):939-945. doi:10.1093/eurheartj/ehu514
I attach the plagiarism report that shows this issue.
[Response] Other than the description for the study design and structure of KNHI database, we used the same definition of modified alcohol intake and descriptions of study results to those of a similar study design [Jeong SM, et al Stroke 2022.]. Nevertheless, we revised the phrases throughout the manuscript to avoid this.
Comment #2. I have some concerns about some issues: Which self-reported questionnaires were used to calculate the frequency of alcohol intake?
[Response] For alcohol intake, the three questions were 1) “how many times do you usually drink per week” and “how many glasses do you drink per occasion”.
We added the description in the Methods section.
“The average frequency of alcohol intake (how many times do you usually drink per week?), and the typical number of standard drinks consumed in a single session (how many glasses do you drink per occasion) were obtained by self-reported questionnaires.”
Comment #3. (In section 2.4: covariates)
1) I think tobacco consumption is briefly described and should have an accurate description.
[Response] We defined cigarette smoking in the Methods section (Covariate)
“Smoking status was categorized into non-smoker, former smoker, and current smoker. The average amount of daily cigarette smoking was also collected.”
2) Although some other covariates are included in the analysis, they are not mentioned in the article, only in Table 1 such as smoking status, physical activity, BMI … I think it would be interesting to correlate them.
[Response] We selected covariates of common risk factors for HF [McMurray JJ. Lancet 2005]. Possible predisposed diseases (hypertension, diabetes mellitus, and dyslipidemia) and their modifiable risk factors of obesity, smoking, and physical activity were included as confounders ) [McManus DD. JACC 2016; Larsson SC et al. JACC 2014; Ruidavets J-B et al. BMJ 2010; Briasoulis A et al. J Clin Hypertens 2012].
We briefly mentioned this in the Methods Section (a brief purpose in Covariates and detailed in Statistical analysis).
“To control for confounders, covariates based on results from the second exam were included in the final analysis.”
“Multivariable models adjusting for age, sex, BMI, smoking status, regular physical activity, area of residence, income, comorbidities (hypertension, diabetes mellitus, and dyslipidemia), systolic blood pressure, and laboratory findings (fasting glucose and total cholesterol) were established.”
References
1) McMurray JJ, Pfeffer MA. Heart failure. Lancet. 2005, 365, 1877-1889.
2) McManus DD, Yin X, Gladstone R, Vittinghoff E, Vasan RS, Larson MG, Benjamin EJ, Marcus GM. Alcohol Consumption, Left Atrial Diameter, and Atrial Fibrillation. J Am Heart Assoc. 2016, 5.
3) Larsson SC, Drca N, Wolk A. Alcohol consumption and risk of atrial fibrillation: a prospective study and dose-response meta-analysis. J Am Coll Cardiol. 2014, 64, 281-289.
4) Ruidavets J-B, Ducimetière P, Evans A, Montaye M, Haas B, Bingham A, Yarnell J, Amouyel P, Arveiler D, Kee F, et al. Pat-terns of alcohol consumption and ischaemic heart disease in culturally divergent countries: the Prospective Epidemiologi-cal Study of Myocardial Infarction (PRIME). BMJ. 2010, 341, c6077.
5) Briasoulis A, Agarwal V, Messerli FH. Alcohol consumption and the risk of hypertension in men and women: a systematic review and meta-analysis. J Clin Hypertens (Greenwich). 2012, 14, 792-798.
Comment #4. (References) Please check the journal´s recommendations, all the references don´t follow the ACS style guide.
[Response] We appreciate reviewer’s comment. We also find that the endnote style for IJEPRH was not formatted in the manuscript of this revision despite the style format was applied in the manuscript which was originally submitted. We think that we will revise references including the citation that are included in this revision in next revision process.
Please see the attachment.

Round 2
Reviewer 2 Report
Dear authors,
I think the authors have improved the manuscript, therefore, I consider that it should be accepted in its present form.
Regards.